# Cortisol and Oxytocin Could Predict Covert Aggression in Some Psychotic Patients

**DOI:** 10.3390/medicina57080760

**Published:** 2021-07-27

**Authors:** Elena Rodica Popescu, Suzana Semeniuc, Luminita Diana Hritcu, Cristina Elena Horhogea, Mihaela Claudia Spataru, Constantin Trus, Romeo Petru Dobrin, Vasile Chirita, Roxana Chirita

**Affiliations:** 1Department of Psychiatry, Faculty of Medicine, “Grigore T. Popa” University of Medicine and Pharmacy, 16th Universitatii, 700115 Iasi, Romania; elenaraduc@yahoo.com (E.R.P.); romeodobrin2002@gmail.com (R.P.D.); d.stigma@gmail.com (R.C.); 2Faculty of Psychology, “Alexandru Ioan Cuza” University of Iasi, Bd. Carol I, 20A, 700505 Iasi, Romania; suzana.semeniuc@yahoo.com; 3Ion Ionescu de la Brad University of Agricultural Science and Veterinary Medicine, 3 Sadoveanu Alley, 700490 Iasi, Romania; rebegeacristina@yahoo.com (C.E.H.); cspataru@uaiasi.ro (M.C.S.); 4Department of Morphological and Functional Sciences, Faculty of Medicine, Dunarea de Jos University, 800008 Galati, Romania; 5Academy of Medical Sciences, 030167 Iasi, Romania; dstigma@gmail.com

**Keywords:** cortisol, oxytocin, covert aggression, psychotic patients

## Abstract

*Background:* The covert or indirect type of aggression has a risk of converting in violent acts and, considering that, it is very important to identify it in order to apply effective preventive measures. In cases of psychotic patients, the risk of becoming violent is harder to predict, as even neuter stimuli may be perceived as threat and trigger aggression. Treating all the psychiatric patients as potential aggressive subjects is not the best preventive measure as only a few of them are aggressive and this measure may further enhance the stigma on mentally ill patients. There is a current need for better understanding of covert aggression and to find objective measures, such as biological markers, that could be indicative of potential violent behavior. In this work, we try to investigate the role of cortisol and oxytocin as potential biomarkers of aggression in patients with psychosis. *Material and Methods:* We analyzed the level of peripheral oxytocin (pg/mL) and cortisol level (ng/mL) in 28 psychotic patients (they were not on psychotropic treatment at the moment of admission and those with substance abuse or personality disorder were excluded from the study) and correlated it with the intensity of aggression reported by the patient (overt and covert type) using the Overt Covert Aggression Inventory and the level of observed aggression of the patient in the past 7 days (rated by the health care provider) using the Modified Overt Aggression Scale. *Results:* We found that psychotic patients with a higher level of covert aggression had a lower level of cortisol (61.05 ± 8.04 ng/mL vs. 216.33 ± 12.6.9 ng/mL, *p* ˂ 0.01) and a higher level of oxytocin (102.87 ± 39.26 vs. 70.01 ± 25.07, *p* = 0.01) when compared with patients with a lower level of covert aggression. Furthermore, we observed significant negative correlation between cortisol and covert aggression (r = −0.676, *p* < 0.001) and between oxytocin and covert type of aggression (r = 0.382, *p* = 0.04). Moreover, we found that a lower level of cortisol together with a higher level of oxytocin are significant predictors of a style of internalized manifestation of aggression, with the predictive model explaining 55% of the variant of the internalized manifestation of aggression (F (2.25) = 17.6, *p* < 0.001, β = 0.35, R^2^ = 55.2). We did not find significant correlations between cortisol and overt aggression, and neither between oxytocin and overt aggression. Positive correlations were also found between the overt type of self-reported aggression and overt aggression reported by the rater (r = 0.459, *p* = 0.01). *Conclusions:* The importance of a predictive model in understanding covert aggression is imperative and the results of our study show that oxytocin and cortisol warrant to be further investigated in establishing a definitive predictive model for covert aggression.

## 1. Introduction

Aggression is a construct that incorporates tendencies or behaviors with the goal of inflicting harm or injury. Actually, aggression is an umbrella term that includes different form of impulses, actions, or behaviors such as verbal aggression, threat, physical violence, property destruction, and even death [1]. Although historically, aggressive behavior had a function as a very important adaptative trait, nowadays, there are fewer instances where this type of behavior is appropriate [2]. Considering that even the covert or indirect type of aggression can lead to negative consequences [3], understanding and correctly identifying various types of aggression is crucial in order to apply effective measures to prevent violent behavior.

Concerning the patients in a psychotic state, the risk of becoming violent is difficult to predict, as neuter stimuli may be perceived as threat and trigger aggressive behavior. For instance, Brucato and his team found that violent ideation was correlated with violent behavior in patients with diagnostic of psychosis [4], but not all psychotic patients are open to revealing their aggressive intent.

Regarding the risk of psychotic patients to become violent, studies have reported mixed results [4,5]. For instance, a Swedish longitudinal study assessed the rates of violence in schizophrenia between 1973 and 2006. They report that 13.2% of individuals suffering from schizophrenia had a record of at least one violent criminal act, compared with 5.3% of individuals in the general population (odds ratio (OR) = 2.0, 95% confidence interval (CI) = 1.8–2.2). The risk of violence was particularly high in schizophrenic individuals with a comorbid substance abuse [6]. Furthermore, Yu et al. reported that mentally ill patients with a personality disorder had a two to three times higher chance of being repeat offenders compared with mentally ill offenders without a personality disorder or offenders without a mental illness [7]. As the studies reported, even when psychosis is not clearly associated with other risk factors known to induce aggression, it seems that there is still a slightly increased chance of becoming violent compared with the general population.

However, treating all the psychiatric patients as potential aggressive subjects is not the best preventive measure. First of all, most of the psychiatric patients are not aggressive and treating them as potential aggressive subjects emphasizes the stigma on mental illness. Secondly, by doing so, we could seriously neglect all other individuals that may pose a real risk of aggressive behavior.

In this way, efforts should be made in better identifying potential aggressive patients. Because the overt type of aggression is easier to observe, the covert or indirect type of aggressive may be difficult to grasp. Patients may hide their aggressive thoughts because of fear of consequences, social condemnation, or the nature of intrusive and ego-dystonic content. Considering that this type of aggression is usually left unmanaged, there is a serious risk for unpredictable violence [8].

Regarding these aspects, it would be useful to rely on objective measure that can give information about the aggressive potential of certain patients. There is a current interest in finding certain biological patterns or biomarkers that could be indicative of potential violent behavior. For instance, some neurotransmitters and hormones have been positively or negatively correlated with increased aggression including serotonin, glutamate, testosterone, cortisol, or oxytocin [9,10,11], but there is not yet a clear understanding of how some biomolecules change in psychotic aggressive patients.

Oxytocin is a well-known molecule for its classical effects in parturition and lactation, with an increased interest lately in its effects in some neuropsychiatric manifestations [12]. Oxytocin was also found to be a modulator for the neuro-endocrine axis, by adjusting the cortisol levels, especially under stress conditions [13]. Thus, oxytocin is mentioned as an anti-stress molecule, in a complex relationship with cortisol and the hypothalamic-pituitary-adrenal axis [14].

In this way, as aggressive behavior can produce devastating consequences and aggressive individuals can be a danger to society, efforts are made in the direction of better identifying potential aggressive patients as well as in applying preventing measures [3]. In this work, we aim to investigate the role of cortisol and oxytocin as potential biomarkers of aggression in patients with psychosis.

## 2. Materials and Methods

The present study enrolled 28 patients (10 males and 18 females) admitted in the psychiatric unit of the Socola Hospital from Iasi, Romania from October to December 2020. Regarding the age of the patients, the average sample was 41.2 (±8.7) years, with the minimum age of 29 years and a maximum age of 59 years. Patients were diagnosed with a psychotic episode according to ICD-10 and DSM V criteria. The subjects were screened for eligibility according to the following inclusion criteria: patients with a psychotic episode, between 18 to 55 years, who were able to read and write. Patients suffering from an endocrinological disorder or other acute medical issues, patients with substance abuse or personality disorder, as well as pregnant or breastfeeding women were excluded from the study. Regarding the confounding variables, we did control the groups for smoking, sex, and age, which could be related to aggression/disease severity. Patients were not on psychotropic treatment at the moment of admission. The Socola Hospital ethical committee approved the present study (no. 12794/17 July 2020) and the patients signed a specific informed approval to be part of this study.

### 2.1. Procedure

The patients were admitted in large hospital rooms, with the possibility to be monitored, with 10 beds inside each room. All patients underwent psychological and psychiatric evaluation and psychometric scales of aggression were administered. The Modified Overt Aggression Scale (MOAS) was administrated by a psychologist, while the Overt-Covert Aggression Inventory (OCAI) was self-administrated.

### 2.2. Aggressiveness Assessment

The Modified Overt Aggression Scale (MOAS) is a hetero-evaluation scale that measures aggression, and is a modified version of the original Overt Aggression Scale. The scale has the advantage of exploring the patient’s aggressive behavior up to seven days prior to application of the scale. The MOAS includes the following four subscales of aggression: verbal, against people, against objects, and self-harm behavior [15]. The scale is rated on a five-point scale of 0 to 4 according to the degree of severity in the different subtypes of aggressive behavior. In each subscale, the score is multiplied by a factor specific for the category, i.e., 1 for verbal aggression, 2 for aggression against objects, 3 for aggression against self, and 4 for aggression against other people. Subscores correlate directly with the degree of severity in the four different domains of aggressive behavior. The instrument showed evidence for validity ant inter-rater reliability (0.94) and reliability test–retest (0.91) [16].

Overt-Covert Aggression Inventory (OCAI) is a brief (10-item) and easy to administer tool that measures self-perception of aggression. OCAI has two subscales, measuring both the overt/direct type of aggression and the covert/indirect type of aggression. Each item is scored on a five-point Likert scale. The internal consistency of the subscales of the Overt-Covert Aggression Inventory using Cronbach coefficient a was measured by the authors, finding a coefficients of 0.73 and 0.74 for the Overt and Obvious Direct Aggression Scale and the Covert and Latent Indirect Aggression Scale, respectively (*n* = 3420: Study 1). The estimates of test–retest reliability of the Overt and Obvious Direct Aggression Scale and the Covert and Latent Indirect Aggression Scale were 0.88 and 0.84, respectively (*n* = 111: Study 2). These results sustain the adequate consistency and reliability of the scale [17]. Permission to use the scale was requested from Dr. MIYAZAKI, Department of Psychosomatic Research. National Institute of Mental Health, National Center of Neurology and Psychiatry, 1-7-3 Kounodai Ichikawa, Chiba, Japan.

Thus, the differences between individuals with high covert aggression (who scored more than 10 on OCAI) (*n* = 14) versus individuals with low levels of covert aggression (with a score of maximum 10 points on OCAI) (*n* = 14) concerning the levels of oxytocin and cortisol were statistically analyzed.

### 2.3. Biochemical Determinations

One serum sample for each patient was obtained by venous blood harvested in the morning, under fasted conditions. The samples were allowed to clot for 2 h at room temperature before centrifugation for 20 min at 1000× *g*. The supernatant was collected and stored at −20 °C before testing.

### 2.4. Oxytocin Measurement

The oxytocin was evaluated using a competitive EIA in 96-well plates: Oxytocin ELISA Kit (LSBio LifeSpan Bio Sciences, Inc., Seattle, WA, USA), following the manufacturer’s instructions. The concentration of oxytocin in the single diluted serum samples (1:10, in PBS) was evaluated with the use of eight standard solutions with well-known oxytocin concentrations in order to generate a standard curve. From the standard stock solution (S1) (1000 pg/mL), standard dilutions were prepared: S2 (500 pg/mL), S3 (250 pg/mL), S4 (125 pg/mL), S5 (62.5 pg/mL), S6 (31.25 pg/mL), S7 (15.63 pg/mL), and S8 (0 pg/mL). This technique used biotinylated detection antibodies, avidin-horseradish peroxidase conjugate (A-HRP-C), and a specific substrate for the enzyme (3,3′,5,5′-Tetramethylbenzidine (TMB)). This substrate develops a blue color with different intensity as a function of the amount of oxytocin in the serum samples. The optical density (OD) value of each well was determined using a microplate reader (Stat Fax 3200 Awareness Technology Inc., Ramsey, MN, USA) set to 450 nm. The OD of the serum sample was compared to the OD of the standard curve generated. In the competition assay, the greater the amount of antigen (oxytocin) in the sample, the lower the color development and optical density reading.

### 2.5. Cortisol Measurement

The cortisol level was evaluated using a competitive inhibition enzyme immunoassay technique: cortisol ELISA Kit (Cusabio Biotech Co., Ltd., Houston, TX, USA), following the manufacturer’s instructions. The concentration of cortisol in the single undiluted serum samples was evaluated with the use of eight standard solutions with well-known cortisol concentrations in order to generate a standard curve. From the standard stock solution (S7) (200 ng/mL), standard dilutions were prepared: S6 (50 ng/mL), S5 (12.5 ng/mL), S4 (3.12 ng/mL), S3 (0.78 ng/mL), S2 (0.195 ng/mL), S1 (0.049 ng/mL), and S0 (0 ng/mL).

This technique used 96-well plates pre-coated with the antigen (cortisol), antibodies specific for cortisol, horseradish peroxidase (HRP) conjugated goat anti-rabbit antibodies, and a specific substrate for the enzyme (3,3′-5,5′-Tetramethylbenzidine—TMB). This substrate develops a blue colour with different intensity as a function of the amount of cortisol in the serum samples. The optical density (OD value) of each well was determined using a microplate reader (Stat Fax 3200 Awareness Technology Inc., Ramsey, MN, USA) set to 450 nm.

The OD of the serum sample was compared to the OD of the standard curve generated. In the competition assay, the greater the amount of antigen (cortisol) in the sample, the lower the colour development and optical density reading. The conversion of the average OD values in ng/mL was accomplished using a formula according to Excel.

## 3. Data Analysis

The differences between individuals with high covert aggression (who scored ˃10 on OCAI) versus individuals with low levels of covert aggression (with a score of maximum 10 points on OCAI) concerning the levels of oxytocin and cortisol were statistically analyzed using Student’s *t*-test (two tailed, unpaired). All results are expressed as mean ± SEM. *p* < 0.05 was regarded as statistically significant. Pearson’s correlation coefficient was used to investigate the possible correlations between the oxytocin and cortisol levels versus the level of aggression (overt or covert). The analyses were performed using the SPSS program (version 20).

## 4. Results

Thus, we investigated the level of oxytocin and cortisol in patients in relation to the intensity of covert type of aggression reported by patients after admission.

We compared the level of oxytocin (pg/mL) as well as cortisol (ng/mL) in patients with a higher level of covert aggression (≥10 points on covert subscale of OCAI scale) and in patients with a lower covert aggression score (≤10 points on covert subscale of OCAI scale) using Student’s *t*-test.

We found that patients with a higher score of covert aggression had a significant lower level of cortisol (61.05 ± 8.04 ng/mL), as compared with patients with a lower score on covert subscale of aggression (*p* < 0.01) (Figure 1).

On the other hand, oxytocin level was significantly higher in patients with a higher score on covert type of aggression (102.87 ± 39.26 pg/mL) compared with patients with a lower score on the covert subscale of OCAI scale (70.01 ± 25.07 pg/mL) (*p* = 0.01) (Figure 2).

We analyzed the association between cortisol and oxytocin levels with the style of manifestation of externalized (overt aggression) versus internalized (covert aggression) aggression, as well as an association between the levels of the two hormones and the aggressive externalized behavior observable at hospitalization.

After applying the Pearson correlations, we observed a statistically significant positive correlation between oxytocin levels and the style of internalized manifestation of aggression (r = 0.382, *p* = 0.04), as well as a negative correlation of cortisol levels and the style of internalized manifestation of aggression (r = −0.676, *p* < 0.001).

Pearson correlation of self-reported overt aggression (OCAI) and aggression reported using the MOAS also indicated a positive significant correlation (r = 0.459, *p* = 0.01).

We were also interested in seeing if cortisol and oxytocin levels could be significant predictors of a style of internalized manifestation of aggression. Thus, after running a stepwise regression analysis, two predictive models emerge. The first considers the level of cortisol as a significant predictor of the style of manifested aggression (F (1.26) = 21.9, *p* < 0.001, β = −0.67, R^2^ = 0.43), so that a low level of cortisol predicts an increased level of internalized manifestation of aggression, which explains 43% of the criterion. Adding the level of oxytocin as a predictor, the new predictive model explains 55% of the variant of the style of internalized manifestation of aggression (F (2.25) = 17.6, *p* < 0.001, β = 0.35, R^2^ = 55.2). We can thus speculate that a low level of cortisol together with a high level of oxytocin predict an increased level of the style of internalized manifestation of aggression.

There was no significant gender difference in oxytocin or cortisol levels, as well as no correlations in this matter with the specific MOAS and OCAI scores.

## 5. Discussion

Psychotic patients may be at risk of aggressive impulses considering the particular stressful state they are in, including having perceptual disturbance and greater thought disorder [18]. Generally, the majority of psychotic patients do not show violent behavior, but there is still an increased risk of their behavior becoming aggressive compared with the general population [19], but the violent burst may be difficult to predict. In a systematic review published by Witt and his team in 2013, violent behavior has been reported to occur in 18.5% of participants with psychosis [20] and the risk of violent attacks is especially increased in association with alcohol consumption or in the case of having a personality disorder [21].

Overt aggression is easier to observe, and thus manage, compared with the internalized type. According to Buss and Durkee (1957). styles of aggression are either covert and latent, indirect aggression, or overt and obvious, direct aggression including verbal and violent behavior [22].

The OCAI scale was used in our study to investigate self-reported overt and covert aggression. Miyzaki and his team correlated the score of the OCAI scale with the Extra-aggression and Intra-aggression Scores of the Picture-Frustration Study and found that people with high covert aggression scores tend to express their anger and aggression to others and repress the anger and aggression toward themselves [23]. This aspect emphasizes the importance of identifying covert aggression considering the high risk of converting to direct aggressive behaviour. The presence of violent thoughts is a risk factor, considering that they can easily become acts, but in many instances, individuals deny having them.

In our study, we observed that patients who reported increased overt aggression also had a higher score on observed overt aggression on MOAS, suggesting that self- perceived aggressive behavior was similar to that observed by an objective rater.

Regarding oxytocin, a classic hormone of love and bonding, it is known to have the opposite effect in the context of threat or exposure to individuals that do not belong to the same group or community [24]. Recently, the role of oxytocin in prosocial or antisocial behavior is framed rather in an interactionist model, by Bartz and his team in 2011, where exogenous or endogenous factors may influence the impact of oxytocin on social behavior. The authors found that oxytocin may increase the perceptual sensitivity to social cues influencing the response to social stimuli [25]. Patients in a psychotic state could interpret neuter environment stimuli as harmful and can develop aggressive thought content against random individuals. In this context, oxytocin may have an influence on how psychotic patients elicit negative responses to social stimuli conditioned by cognitive distortion. Moreover, it was previously demonstrated that, in an unfamiliar environment, oxytocin could generate hostile behavior [26].

In our study, we found that elevated oxytocin is positively associated with the internalized type of aggression, but not with the overt type. Studies found a dysregulated oxytocinergic system in schizophrenic patients that is thought to be related to perceptual and social disturbances, especially emotional face perception [27,28]. A higher oxytocin level in psychotic patients may increase attention to social cues, while the disturbed perception may determine negative interpretation of neuter or positive social interaction.

In our study, we also found that oxytocin, together with cortisol, can predict an increased risk of covert aggression.

Glucocorticoids and HPA axis play a significant role in aggression [29]. Studies have found that, in healthy individuals, an acute increase in the cortisol level increases reactive aggression [30]. A lower cortisol level is found in pathological forms of aggression (proactive aggression), which may be explained by HPA axis alterations. For instance, a study found that, in extremely aggressive boys, there is a persistent lower salivary cortisol finding [31].

Moreover, it appears that, even in psychotic patients, a lower level of glucocorticoid was linked to abnormal forms of aggressive behavior by Das and his group. They compared the modification of cortisol during the day in aggressive patients, psychotic and non-psychotic, as well as non-aggressive individuals (psychotic and non-psychotic). The authors found lower levels of morning cortisol, afternoon cortisol, and cortisol variability among the aggressive group (vs. non aggressive group) and among the psychotic group versus the non-psychotic group. These results show that the lower cortisol level may represent a biological status of aggression independent of the presence of psychosis [32].

In our study, we investigated the relation between cortisol and two forms of aggression, the covert and overt type, and found that individuals with a higher internalized aggression had a lower cortisol level compared with individuals with a lower internalized aggression. There is some evidence that patients with psychosis or long before developing psychosis have a higher production of cortisol, which may contribute to HPA dysregulation by suppressing HPA axis, explaining the low cortisol level in aggression [33,34].

In addition, in our study, we also found a lower level of cortisol as a predictor of covert type of aggression, suggesting that biological factors such as cortisol and oxytocin might be mediators in internalized aggression and potential useful markers in identifying this type of aggression.

We could also mention that previous reports described decreased levels of cortisol and higher peripheral oxytocin in response to stress in individuals with early trauma—some of them with dissociative disorder [35,36]. However, they did not specifically differentiate the matters between overt and covert aggression.

Regarding the limitations of our study, we can mention the fact that we did use unextracted serum samples, considering the possible affected reliability of non-extracted diluted plasma/serum samples for oxytocin measurement [37], as well as using ELISA in single, considering that a single cell is less reliable than a more standard duplicate measure. We can also mention here that the MOAS and OCAI scale were only translated in Romanian, but not re-validated. Another important limitation can be represented by the fact that cortisol was also measured once, rather than a more complex multiple time-points analysis, as it is known to vary considerably throughout the day depending on a variety of other factors [38].

## 6. Conclusions

The importance of a predictive model in understanding covert aggression is imperative and the results of our study show that oxytocin and cortisol warrant to be further investigated in establishing a definitive predictive model for covert aggression.

## Figures and Tables

**Figure 1 medicina-57-00760-f001:**
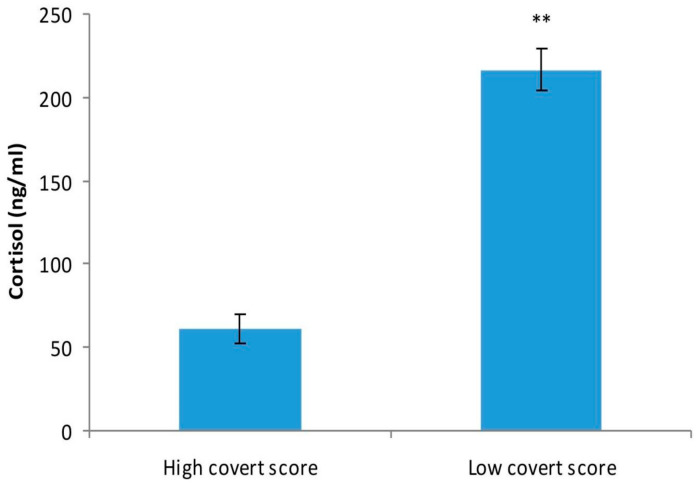
Cortisol level (ng/mL) in patients with a higher level of covert aggression and patients with a lower covert aggression score, as studied by two-tailed, unpaired Student’s *t*-test. The values are mean ± S.D. (*n* = 14 per group). ** *p* < 0.01.

**Figure 2 medicina-57-00760-f002:**
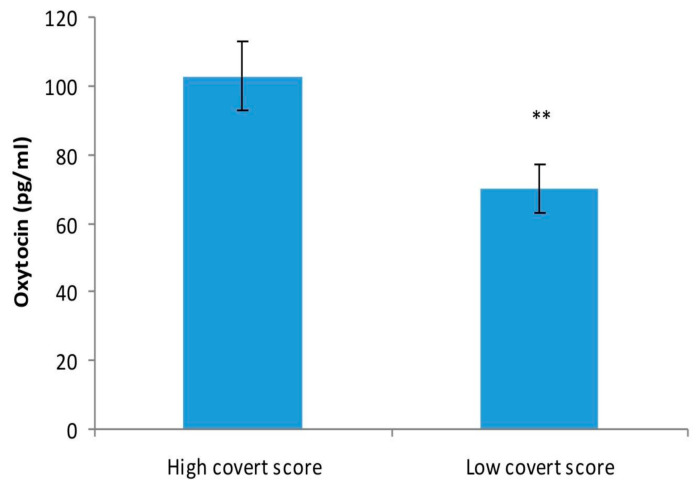
Oxytocin level (pg/mL) in patients with a higher level of covert aggression and patients with a lower covert aggression score, as studied by two-tailed, unpaired Student’s *t*-test. The values are mean ± S.D. (*n* = 14 per group). ** *p* = 0.01.

## Data Availability

All raw data is available on request.

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
