# Peer review of "Cortisol and Oxytocin Could Predict Covert Aggression in Some Psychotic Patients"

_medicina, 2021, doi:10.3390/medicina57080760_

Round 1
Reviewer 1 Report
The authors presented a manuscript about the discovery of possible objective biomarkers of aggression in psychotic patients. They investigated the link between covert and overt aggression and serum levels of cortisol and oxytocin. Possible findings can be interesting for future clinical guidelines for psychiatrists as well as for basic research of aggression and the role of neuropeptides and steroids in this process. However, before publishing several points should be clarified within the manuscript.
1) Criteria of inclusion is Psychotic episode. Authors discussed in the introduction different research regarding psychotic patients and aggression including patients with schizophrenia. However in the proposed design only persons with psychotic episodes included (Which I believe F23 in ICD-10). Justify, why this group specifically.
2) Authors mentioned that patients who suffered from illegal drug abuse were excluded. But what about antipsychotics and other psychotropic medications? The authors indicated they did control, but no data presented.
3) Authors included 10 males and 18 females. But presented data is pooled. There any gender difference in oxytocin or cortisol levels as well as correlation with MOAS and OCAI scores? We can expect some difference in serum oxytocin levels. Therefore, it is interesting to know.
4) I think it is important to mention in discussion as limitations and open questions. Elevated risk of violent behavior observed in patients comorbid with substance abuse and/or personality disorders. However current research excludes these comorbid groups: Can other factors related to comorbidity more important for psychotic patients? How about cortisol and oxytocin levels in patients with substance abuse and personality disorder? Also how much reliable will be proposed model will be in that case. It is possible that levels of cortisol and oxytocin will not predict covert aggression in those high-risk groups.
5) I did not find data regarding samples on plates for cortisol and oxytocin ELISA. As the single, duplicated, triplicated?
6) Also, LSBio LifeSpan ELISA kit required to test the dilution of samples. Did the authors' used non-diluted serum samples, or diluted, please indicate.
7) Please carefully check English and typos. A lot of different typing of the same words. For example oxytocin/oxytocine; cover/covert; p/ml. Sometimes words not in English (à jeun)
8) I quite confused with Figure 2. SEM for groups with n = 14 quite high. I cannot reproduce t-test value p=0.01 with this data. Are you sure it is not SD? Also, it seems some diversity in oxytocin levels exists. It will be very representative to make figures 1 and 2 with individual points for each sample.
9) It will be good to clarify the conditions of patients during experiments somewhere (in methods for example) they were in the hospital? Isolated or inwards with other patients. Cortisol and oxytocin levels are very sensitive to stress and group/isolation conditions.
10) Figure 3-5 with correlation analysis. In all three figures, 14 points are presented. However the number of patients – is 28. Why only half presented?
11) According to methods, authors carefully followed the instruction for LSBio LifeSpan ELISA for oxytocin, and that instruction did not require extraction of plasma/serum. However, there a never-ending scientific debate about the reliability of unextracted diluted plasma/serum samples for oxytocin measurement. Therefore it will be nice to mention this problem in the discussion and justify the choice of using non-extracted serum samples. Also, we can see it in the presented data. Range of measured oxytocin 50-150 pg/ml wherein extracted samples it’s usually 1-10 pg/ml.
Author Response
Reviewer #1:
The authors presented a manuscript about the discovery of possible objective biomarkers of aggression in psychotic patients. They investigated the link between covert and overt aggression and serum levels of cortisol and oxytocin. Possible findings can be interesting for future clinical guidelines for psychiatrists as well as for basic research of aggression and the role of neuropeptides and steroids in this process. However, before publishing several points should be clarified within the manuscript.
1) Criteria of inclusion is Psychotic episode. Authors discussed in the introduction different research regarding psychotic patients and aggression including patients with schizophrenia. However in the proposed design only persons with psychotic episodes included (Which I believe F23 in ICD-10). Justify, why this group specifically.
Response: We were only interested in this aspect because we were interested only in the patients in the acute phase, in order to investigate patients without treatment. Thank you!
2) Authors mentioned that patients who suffered from illegal drug abuse were excluded. But what about antipsychotics and other psychotropic medications? The authors indicated they did control, but no data presented.
Response: We added the following details as you kindly requested: “Patients suffering from an endocrinological disorder or other acute medical issues, patients with substance abuse or personality disorder, as well as pregnant or breastfeeding women were excluded from the study. Regarding the confounding variables, we did control the groups for smoking, sex and age which could be related to aggression/disease severity. Patients were not on psychotropic treatment at the moment of admission.”
3) Authors included 10 males and 18 females. But presented data is pooled. There any gender difference in oxytocin or cortisol levels as well as correlation with MOAS and OCAI scores? We can expect some difference in serum oxytocin levels. Therefore, it is interesting to know.
Response: There were no significant differences between sexes in hormones levels, as well as in studying the correlations with the scales. We added a phrase on this matter on the Results section now.
4) I think it is important to mention in discussion as limitations and open questions. Elevated risk of violent behavior observed in patients comorbid with substance abuse and/or personality disorders. However current research excludes these comorbid groups: Can other factors related to comorbidity more important for psychotic patients? How about cortisol and oxytocin levels in patients with substance abuse and personality disorder? Also how much reliable will be proposed model will be in that case. It is possible that levels of cortisol and oxytocin will not predict covert aggression in those high-risk groups.
Thank you! The reason we did exclude personality disorder comorbidities and substance abuse is the fact that the previous studies (. S. Fazel, N. Langström, A. Hjern, M. Grann, and P. Lichtenstein: Schizophrenia, substance abuse, and violent crime. The Journal of the American Medical Association. 301(19): 2016–23. 2009.///// Yu R, Geddes JR, Fazel S: Personality disorders, violence, and antisocial behavior: a systematic review and meta-regression analysis. J Pers Disord. 26:5. 2012.) already demonstrated an increase in aggression in psychotic patients with comorbidities and substance abuse.
5) I did not find data regarding samples on plates for cortisol and oxytocin ELISA. As the single, duplicated, triplicated?
Details were added. Thank you.
6) Also, LSBio LifeSpan ELISA kit required to test the dilution of samples. Did the authors' used non-diluted serum samples, or diluted, please indicate.
Details were added. Thank you.
7) Please carefully check English and typos. A lot of different typing of the same words. For example oxytocin/oxytocine; cover/covert; p/ml. Sometimes words not in English (à jeun).
We did replace and corrected. Thank you!
8) I quite confused with Figure 2. SEM for groups with n = 14 quite high. I cannot reproduce t-test value p=0.01 with this data. Are you sure it is not SD? Also, it seems some diversity in oxytocin levels exists. It will be very representative to make figures 1 and 2 with individual points for each sample.
It is indeed SD. Thank you! We did correct it.
9) It will be good to clarify the conditions of patients during experiments somewhere (in methods for example) they were in the hospital? Isolated or inwards with other patients. Cortisol and oxytocin levels are very sensitive to stress and group/isolation conditions.
We added that “The patients were admitted in large hospital rooms, with possibility to be monitored, with 10 beds inside each room”.
10) Figure 3-5 with correlation analysis. In all three figures, 14 points are presented. However the number of patients – is 28. Why only half presented?
It was an inseration of data error by our side. We are sorry about that. Still, as requested by the second reviewer, the Figures 3 to 5 were now removed from the manuscript, as they were considered exhaustive.
11) According to methods, authors carefully followed the instruction for LSBio LifeSpan ELISA for oxytocin, and that instruction did not require extraction of plasma/serum. However, there a never-ending scientific debate about the reliability of unextracted diluted plasma/serum samples for oxytocin measurement. Therefore it will be nice to mention this problem in the discussion and justify the choice of using non-extracted serum samples. Also, we can see it in the presented data. Range of measured oxytocin 50-150 pg/ml wherein extracted samples it’s usually 1-10 pg/ml.
Thank you! We did add this aspect now, as well as a relevant reference in this matter.

Reviewer 2 Report
In an effort to find preventive and objective measures, such as biological markers, that could be indicative of potential violent behavior in psychotic hospitalized patients, the authors examine the potential role of cortisol and oxytocin as biomarkers of aggression in such patients.
The main result is in some way counter-intuitive: morning measures of cortisol (dubbed as "stress hormone") is lower in patients with a high self-evaluation of covert aggression, whereas peripheral oxytocin (dubbed as "love hormone") is higher in the same patients.
The objectives of the study are relevant, method adequate (note that there is a satisfactory association between self- and hetero-evaluation of aggression) ; biological measures are adequate, analysis are relevant, and results deserve to be published because, precisely, they question the mainstream belief that oxytocin is a « pacific » hormone, whereas many studies already seem to underline its association with aggression, in some circumstances. Therefore, this study opens to a range of theoretical questions as well as practical suggestions in the clinical practice. The paper is clear, well written.
Some remarks :
- At my sense, the counter-intuitive aspect of the results deserve to be more addressed. Already in the introduction, when the authors say «…thus, oxytocin is mentioned as an anti-stress molecule…», so what were they expected before running the study? In the conclusion, they say appropriately that «… oxytocin may have an influence on how psychotic patient elicit negative responses to social stimuli conditioned by cognitive distortion. ». This is interesting, however the question remain why these patients have higher levels of oxytocin.
- Chapter 2.1, Aggressiveness assessment : precise that MOAS is an hetero-evaluation.
- End of the same chapter: correct « …cover aggression »
- Precise : these scales MOAS and OCAI : have they been translated into Romanien ? And not re-validated ?
- Chapter 2.2. Biochemical determinations : a unique morning sample has been collected for each patient ?
- I think that a chapter « Procedure » would have been suitable : how many patients accepted/refused to participate, how long had they been hospitalized, who administered the MOAS, etc.
- Chapter Results : authors use that terminology : « …the style of manifestation of aggression externalized vs. Internalized », which has not been introduced before. Is it synonymous with overt and covert aggression ?
- Figures 3 and 4 : not usefu
- Rather than these figures 3 and 4, we would be interested in the correlations between overt aggression and oxytocin (non-sign.) and between overt and cortisol (sign.), as mentioned later in the discussion (see sentence in discussion : « In our study we found that elevated oxytocin is positively associated with an internalized type of aggression but not with the overt type »).
- Figure 5: not useful
- Concerning the step-wise regression analysis, we would be interested to know which variables have been introduced in the model.
- Verify the sentence «…predictor of the style of manifested manifestation of aggression… »
- Note that other teams found lower cortisol and higher peripheral oxytocin in response to stress in individuals with early trauma, some of them with dissociative disorder, e.g. Dimitrova, N. et al. Closeness in Relationships as a Mediator Between Sexual Abuse in Childhood or Adolescence and Psychopathological Outcome in Adulthood. Clinical Psychology and Psychotherapy, 17, 183–195 (2010); Pierrehumbert, B. et al. Adult attachment representations predict cortisol and oxytocin responses to stress. Attachment & Human Development, Vol. 14, No. 5, September 2012, 453–476.
Author Response
Reviewer 2:
In an effort to find preventive and objective measures, such as biological markers, that could be indicative of potential violent behavior in psychotic hospitalized patients, the authors examine the potential role of cortisol and oxytocin as biomarkers of aggression in such patients.
The main result is in some way counter-intuitive: morning measures of cortisol (dubbed as "stress hormone") is lower in patients with a high self-evaluation of covert aggression, whereas peripheral oxytocin (dubbed as "love hormone") is higher in the same patients.
The objectives of the study are relevant, method adequate (note that there is a satisfactory association between self- and hetero-evaluation of aggression) ; biological measures are adequate, analysis are relevant, and results deserve to be published because, precisely, they question the mainstream belief that oxytocin is a « pacific » hormone, whereas many studies already seem to underline its association with aggression, in some circumstances. Therefore, this study opens to a range of theoretical questions as well as practical suggestions in the clinical practice. The paper is clear, well written.
Thank you very much!
Some remarks :
At my sense, the counter-intuitive aspect of the results deserve to be more addressed. Already in the introduction, when the authors say «…thus, oxytocin is mentioned as an anti-stress molecule…», so what were they expected before running the study? In the conclusion, they say appropriately that «… oxytocin may have an influence on how psychotic patient elicit negative responses to social stimuli conditioned by cognitive distortion. ». This is interesting, however the question remain why these patients have higher levels of oxytocin.
Response: We did add that “Patients in a psychotic state could interpret neuter environment stimuli as harmful and can develop aggressive thought content against random individuals. In this context, oxytocin may have an influence on how psychotic patient elicit negative responses to social stimuli conditioned by cognitive distortion. Also, it was previously demonstrated that in an unfamiliar environment oxytocin could generate hostile behavior (Duque-Wilckens N, Steinman MQ, Busnelli M, Chini B, Yokoyama S, Pham M, Laredo SA, Hao R, Perkeybile AM, Minie VA, Tan PB, Bales KL, Trainor BC. Oxytocin Receptors in the Anteromedial Bed Nucleus of the Stria Terminalis Promote Stress-Induced Social Avoidance in Female California Mice. Biol Psychiatry. 2018 Feb 1;83(3):203-213.)”
Chapter 2.1, Aggressiveness assessment: precise that MOAS is an hetero-evaluation.
Response: We did add this aspect. Thank you!
End of the same chapter: correct « …cover aggression »
Response: We did correct. Thank you!
Precise : these scales MOAS and OCAI : have they been translated into Romanien ? And not re-validated ?
Response: Yes. That is correct. Thank you. We did add a phrase on the limitations section on this matter.
Chapter 2.2. Biochemical determinations : a unique morning sample has been collected for each patient ?
Response: Yes. We did add that detail now. Thank you.
I think that a chapter « Procedure » would have been suitable : how many patients accepted/refused to participate, how long had they been hospitalized, who administered the MOAS, etc.
Response: We did add these details about the scales. Regarding the refusing patients, we did not have any refusing patient. Regarding the days of hospitalization, we administrated the MOAS scale at admission and OCAI scale 7 days after admission. We did not follow the patients after this period. Thank you.
Chapter Results : authors use that terminology : « …the style of manifestation of aggression externalized vs. Internalized », which has not been introduced before. Is it synonymous with overt and covert aggression ?
Response: Yes. We did explain now. Thank you.
Figures 3 and 4 : not useful
Rather than these figures 3 and 4, we would be interested in the correlations between overt aggression and oxytocin (non-sign.) and between overt and cortisol (sign.), as mentioned later in the discussion (see sentence in discussion : « In our study we found that elevated oxytocin is positively associated with an internalized type of aggression but not with the overt type »).
Figure 5: not useful
Response: We did delete Figures 3-5 now. Thank you.
Concerning the step-wise regression analysis, we would be interested to know which variables have been introduced in the model.
Response: The predictors introduced in the regression analysis were cortisol and oxytocin levels. The analysis excluded oxytocin levels as a significant single predictor and two significant predictive patterns remained: cortisol levels and cortisol + oxytocin levels.
Verify the sentence «…predictor of the style of manifested manifestation of aggression… »
Response: We did corect now. Thank you.
Note that other teams found lower cortisol and higher peripheral oxytocin in response to stress in individuals with early trauma, some of them with dissociative disorder, e.g. Dimitrova, N. et al. Closeness in Relationships as a Mediator Between Sexual Abuse in Childhood or Adolescence and Psychopathological Outcome in Adulthood. Clinical Psychology and Psychotherapy, 17, 183–195 (2010); Pierrehumbert, B. et al. Adult attachment representations predict cortisol and oxytocin responses to stress. Attachment & Human Development, Vol. 14, No. 5, September 2012, 453–476.
Response: Thank you. We added these aspects in the discussion section now : ‘Also we could mention that previous reports described decreased levels of cortisol and higher peripheral oxytocin in response to stress in individuals with early trauma -some of them with dissociative disorder (37, 38). However, they did not specifically dif-ferentiate the matters between over and covert aggression.”
Kind regards-The Authors.
Round 2
Reviewer 1 Report
The authors did a good job and responded to each question of review. Now presented data looks more clear for understanding and can be interesting for readers. My suggestions:
1: to check references and correct them in the right numerical order (after modification of manuscript new references sometimes in not correct order).
2: Authors indicate they measure cortisol and oxytocin by ELISA in single. Please add this also in your description of limitations cause a single cell is less reliable than a more standard duplicate measure.
Some typos I believe can be checked and corrected by editors during proofreading.
Author Response
Response letter
Dear Editor-in-Chief,
Thank you for your reply on our manuscript. We appreciated the valuable comments of the reviewer. Here is a list of all the changes we made and our responses to the reviewers' suggestions:
Reviewer #1:
The authors did a good job and responded to each question of review. Now presented data looks more clear for understanding and can be interesting for readers.
Thank you for helping us in improving the quality of our manuscript.
My suggestions:
1: to check references and correct them in the right numerical order (after modification of manuscript new references sometimes in not correct order).
We did correct now. Thank you and please accept our apologies for the mistake.
2: Authors indicate they measure cortisol and oxytocin by ELISA in single. Please add this also in your description of limitations cause a single cell is less reliable than a more standard duplicate measure.
Some typos I believe can be checked and corrected by editors during proofreading.
We did add now this aspect in the Limitations part, at the end of the Discussion section. Typos were also searched and corrected though the manuscript. Thank you!
Kind regards, The Authors.
